# Unveiling the Chemical Profiling Antioxidant and Anti-Inflammatory Activities of Algerian *Myrtus communis* L. Essential Oils, and Exploring Molecular Docking to Predict the Inhibitory Compounds against Cyclooxygenase-2

**DOI:** 10.3390/ph16101343

**Published:** 2023-09-22

**Authors:** Samia Belahcene, Widad Kebsa, Damilola A. Omoboyowa, Abdulaziz A. Alshihri, Magbool Alelyani, Youssef Bakkour, Essaid Leghouchi

**Affiliations:** 1Laboratory of Biotechnology, Environment and Health, Faculty of Nature and Life Sciences, University of Jijel, Jijel 18000, Algeria; 2Laboratory of Molecular Toxicology, Faculty of Nature and Life Sciences, University of Jijel, Jijel 18000, Algeria; w.kebsa@univ-jijel.dz; 3Laboratory of Phyto-Medicine and Computational Biology, Department of Biochemistry, Adekunle Ajasin University, Akungba Akoko 342111, Ondo State, Nigeria; 4Department of Radiological Sciences, College of Applied Medical Science, King Khalid University, Abha 61421, Saudi Arabia

**Keywords:** *Myrtus communis* L., essential oils, bioactive compounds, antioxidant, anti-inflammatory, cyclooxygenase-2

## Abstract

Considering the large spectrum of side effects caused by synthetic drugs and the development of natural alternatives utilizing Algerian flora, this study aimed to place a spotlight on the chemical profile and antioxidant and anti-inflammatory activities of *Myrtus communis* L. essential oils (MCEOs). In this study, essential oils (EOs) were collected via hydro-distillation of the plant’s leaves, and a chemical constituent analysis was performed using gas chromatography–mass spectrophotometry (GC–MS). The in vitro antioxidant activity was evaluated using DPPH, ABTS, and hydroxyl radical scavenging tests. The in vitro anti-inflammatory capacity was estimated by studying the antidenaturation effect using bovine serum albumin (BSA) as a protein model. The in vivo anti-inflammatory activity was carried out by utilizing the classical model of carrageenan-induced paw edema in rats, using diclofenac (DCF) as the reference drug. Moreover, the molecular interaction of the compounds obtained from the GC–MS analysis was studied within the binding site of cyclooxygenase-2 (COX-2) using an in silico approach as the confirmatory tool of the in vitro and in vivo experiments. The GC–MS analysis revealed that MCEOs were mainly composed of oxygenated monoterpenes (70.56%), oxygenated sesquiterpenes (3.1%), sesquiterpenes (4.17%), and monoterpenes (8.75%). Furthermore, 1,8-cineole was the major compound (19.05%), followed by cis-geranyl acetate (11.74%), methyl eugenol (5.58%), α-terpineol (4.62%), and β-myrcene (4.40%). MCEOs exhibited remarkable concentration-dependent free radical scavenging activity, with an IC_50_ of 15.317 ± 0.340 µg/mL, 18.890 ± 2.190 µg/mL, and 31.877 ± 0.742 µg/mL for DPPH, ABTS, and hydroxyl radical, respectively. The significant in vitro anti-inflammatory activity due to the inhibition of BSA denaturation was proportional to the EO concentration, where the highest value was recorded at 100 μg/mL with an approximately 63.35% percentage inhibition and an IC_50_ of 60.351 ± 5.832 μg/mL. MCEOs showed a good in vivo anti-inflammatory effect by limiting the development of carrageenan-induced paw thickness. The in silico study indicated that, among the 60 compounds identified by the GC–MS analysis, 9 compounds were observed to have a high binding energy to cyclooxygenase-2 as compared to diclofenac. Our study revealed that EOs from Algerian *Myrtus communis* L. can be considered to be a promising candidate for alleviating many debilitating health problems and may provide new insights in the fields of drug design, agriculture, and the food industry.

## 1. Introduction

Although the world is now becoming techno-savvy in order to improve living standards, people are still returning to nature to seek solutions for a healthy life [1,2,3]. Recently, the World Health Organization estimated that 80% of people, notably in developing societies, used traditional medicines in their daily routine, which proves that plants and herbs can offer harmless and subtle ways to promote the human lifestyle [4]. The renewed interest in medicinal plants is based on their ubiquity and their pool of dynamic ingredients that could help the body in defending itself against different injuries caused by oxidative stress [5].

Free radicals are important mediators that initiate oxidative stress and inflammation when they are not maintained at low physiological concentrations [6]. However, the human body regulates this unbalanced redox state by employing endogenous antioxidants as the first line of defense [6,7]. When ROS production persists, it may overwhelm the endogenous capacity, leading to a failure in scavenging and neutralizing reactive species. This can be translated as a disruption of basic biomolecules in cells, notably proteins, lipids, and genetic materials [8,9]. It is worth noting that inflammation is an indispensable immune response to tissue injury, involving a complex cascade of reactions necessary to repair damage, eliminate invading pathogens, and maintain adequate homeostasis [10]. When acute inflammation is not resolved, it may become chronic, leading to a vicious circle connecting inflammation and its associated pathophysiological scenarios [11], like hyper cell stimulation, inappropriate cell death, progressive organ damage, or the genesis of different cancers, and it promotes all stages of tumorigenesis [12,13].

In this regard, many synthetic anti-inflammatory and antioxidant agents have been developed to remediate oxidative stress and inflammation [14,15]. NSAIDs are known to block the biosynthesis of prostaglandins (PGs) from arachidonic acid and thereby suppress the upstream production of cyclooxygenases (COX-1 and -2) [16]. Unfortunately, several studies have vividly explained that the rampant, indiscriminate, and long-term use of manufactured drugs is coupled with various devastating negative effects on nontarget tissues in a silent way [11,17], affecting kidney function [18], liver metabolism [19], and the gastrointestinal tract [16,20]. Likewise, many factors remain as major drawbacks in combating pathophysiological diseases with synthetic drugs, such as their low efficiency, a lack of availability, the prohibitive cost, and the emergence of drug-resistant strains [14,20,21,22]. The failure of these specific candidate drugs and a lack of promising treatments presents a worrisome problem that raises the question of whether switching to green alternative discoveries is the best solution [23,24]. In a continuous effort to search for novel alternatives that are free of side effects and independent of any age groups or sexes to prevent, manage, and/or treat human disorders, especially chronic diseases, several scientists have shifted their main focus towards EOs as a new approach over synthetic drugs [2]. Currently, there is a paucity of robust data on the behavior of these volatile compounds, and scientific evidence underpinning their safety and efficacy is seriously needed [25]. That is why EOs, in particular, are in need of rigorous scientific research.

Myrtle (*Myrtus communis* L., Myrtaceae) is an aromatic plant with a medicinal reputation and a consolidated ethno-botanical tradition [26]. It is an evergreen sclerophyll shrub, commonly known in Algeria as Al-Rihan [27]. It is widespread in Mediterranean regions, where it grows and is scattered spontaneously [28]. In folk medicine, myrtle leaves and fruits have been used as a decoction and an infusion and as a health remedy for a large spectrum of pathologies. Myrtle extracts have been deeply documented for their distinguished biological values [29], but essential oils remain underexplored in our region. MCEOs are an inexhaustible source of biologically active substances, which may demonstrate versatile ethno-pharmacological activities such as antimicrobial [30,31], antioxidant [32], antidiabetic [33], anticancer [34], and anti-inflammatory effects [35,36]. The attempts to comprehensively undertake in vitro/in vivo/in silico analyses of EOs at the molecular level, including target enzymes, are lacking, and this needs to be examined in depth. 

In order to search for something new, natural, and safe, with proven scientific effectiveness and new dimensions of biocompatibilities, our study intended to unravel the phytochemical profile of MCEOs, their antioxidant activity, and the in vitro, in silico, and in vivo anti-inflammatory effects using a model of acute inflammation provoked by an acute dose of carrageenan in rats. As far as we are aware, this is the first reliable report devoted to the knowledge of the mechanism underlying the molecular interactions and affinities of MCEOs with key proteins, in an important inflammation-related pathway of arachidonic acid, using molecular docking analysis.

## 2. Results

### 2.1. Extraction Yield of MCEOs

Hydro-distillation was used to extract the essential oil (EO) from *Myrtus communis* L. in the balsamic period. The EO yield was estimated based on an averaged oil weight/dry weight, as reported in Table 1.

### 2.2. Chemical Profile of EOs from Myrtus communis L.

Hydro-distillation of *M. communis* leaves gave a golden-yellow EO that emitted an aromatic and refreshing smell with a very intense, quite unpleasant, and strongly bitter taste. GC–MS was performed to determine the composition of bioactive compounds and the chemotype of the myrtle essential oil. Altogether, 60 compounds, which represented 98.78% of the total oil, were quantified (Figure 1, Table 2). 

Oxygenated monoterpenes (OMs) were the main compounds (70.56%), followed by monoterpene hydrocarbons (8.75%), sesquiterpene hydrocarbons (4.17%), and oxygenated sesquiterpenes (OSs) (3.10%). The major detected constituents were as follows: 1,8-cineol (19.05), cis-geranyl acetate (11.74%), methyl eugenol (5.58%), α-terpineol (4.62%), and β-myrcene (4.40%). Other compounds were detected in lower amounts, with values lower than 3.0% or as traces. Our oil sample was characterized by the presence of five monoterpene hydrocarbons (MHs), four sesquiterpenes hydrocarbons (SHs), forty-seven OMs, and six OSs, in addition to other molecules. The MHs represented 8.75%, the SHs 4.17%, the OMs 83.65%, the OSs 3.1%, and the remaining compounds represented 0.3%. Table 3 illustrated the relative composition of the identified compounds classified according to the functional groups. 

### 2.3. Antioxidant Activity

Several essential oils have antioxidant properties, and their use as natural antioxidants is gaining popularity, like some synthetic antioxidants (such as BHA and BHT), which are questionable due to potential health risks [37]. 

In this study, different concentrations of MCEOs were tested for their DPPH, ABTS, and hydroxyl scavenging activities. This was in comparison with gallic acid as the positive control, which has a very well-known antioxidant power (Figure 2). A strong association between concentration and antioxidant activities was observed; our oil sample scavenged radicals in a concentration-dependent manner in all assays. The percentage of DPPH inhibition increased from 11.52 ± 0.96% at 2.5 µg/mL to 75.79 ± 0.56% at 40 µg/mL (Figure 2a). The inhibition percentage ranged from 6.24 ± 1.78% to 70.41 ± 0.37% in the ABTS test (Figure 2c). 

Hydroxyl radical was scavenged by MCEOs with an inhibition percentage ranging between 2.14 ± 1.78 and 60.85 ± 5.03% (Figure 2e). The smallest concentration of MCEOs required for 50% inhibition (IC50 values) was 15.317 ± 0.340 µg/mL, 18.890 ± 2.190 µg/mL, and 31.877 ± 0.742 µg/mL for DPPH, ABTS, and hydroxyl radical, respectively. The IC50 of gallic acid was 13.188 ± 0.223, 16.887 ± 0.863, and 33.217 ± 1.805 µg/mL with DPPH, ABTS, and hydroxyl radical, respectively (Figure 2b,d,f). A lower IC50 means that the compound under study is more effective. The MCEOs showed a strong antioxidant activity when comparing IC50s. Our tested oil presented the same effect (*p* > 0.05) as GA in ABTS and OH• tests.

### 2.4. Anti-Inflammatory Activity

#### 2.4.1. In Vitro Study: Inhibition of BSA Denaturation 

MCEOs were analyzed for their anti-inflammatory activity using the albumin denaturation assay. The results are summarized in Figure 3a. A strong association between concentration and inhibition percentage of BSA denaturation was observed. Indeed, the effect of MCEOs against the heat denaturation of BSA was found to be approximately threefold lower than that of sodium diclofenac. These observations were confirmed by comparing their IC50 values. The ability of our EO sample to conserve the BSA from heat-induced denaturation is inextricably linked to concentration and reflected by an IC50 value of 60.351 ± 5.832 µg/mL as compared to that of standard sodium diclofenac at 7.110 ± 0.624 µg/mL (Figure 3b). At the highest tested concentration (100 μg/mL) at which diclofenac almost completely inhibited the denaturation process (99.4 ± 0.34%), the MCEOs showed an anti-inflammatory activity at a percentage inhibition level of 61.80 ± 1.83 (%). The oil holds exceptional effect, being able to preserve BSA from denaturation by approximately 47.57 ± 1.83 (%) at (50 µg/mL) and by 34.88 ± 0.94% at (25 µg/mL). At a concentration of 12.5 µg/mL, MCEOs were observed to be efficacious by significantly (*p* < 0.001) inhibiting heat-induced denaturation of BSA by 21%, compared to the control. This proves that our oil sample, even at low concentrations, is still effective. The MCEOs tested in our experiments exhibited a capacity to protect BSA against denaturation.

#### 2.4.2. In Vivo Study: Carrageenan-Induced Paw Edema in Rats

The model of carrageenan (CAR)-induced inflammation in rats was used to study the anti-inflammatory activity of MCEOs. Sub-plantar injection of 1% CAR to the control group revealed a significant (*p* < 0.05) increase in the percentage augmentation of paw diameter. This is evidence that CAR injection into the rats’ paws resulted in acute inflammation. The mechanism of action of DCF has been well studied; therefore, it was used as the standard drug. The validity and reliability of the results from this study were also improved by providing a reference for the anti-inflammatory efficacy of the tested EO. During the first hour, the reference drug and the MCEOs at two dose levels did not inhibit paw edema. The percentage augmentation was 89.22%, 83.88%, and 89.49% in the rats that received DCF, MCEOs 50 mg/kg, and MCEOs 25 mg/kg, respectively. In the fourth hour after induction with carrageenan, the augmentation percentage was significantly decreased to 28.49%, 28.71% and 39.65%, and with the standard drug and the two dose levels (25 and 50 mg/kg) of MCEOs, respectively. 

In this study, our oil sample significantly stopped the progression of edema at 50 and 25 mg/kg within 4 h (Figure 4a). The MCEOs showed the same inhibition as diclofenac (*p* > 0.05). Figure 4a shows the effect of MCEOs on the effect of carrageenan-induced edemas on the posterior paw of rats relative to time. In this study, the administration of DCF (50 mg/kg bodyweight) significantly reduced the increase in paw edema. The anti-inflammatory effect of MCEOs was optimal 3 h after the induction of the edema. At 25 mg/kg, MCEOs resulted in significant anti-inflammatory activity that was similar (*p* > 0.05) to that of DCF at (50mg/kg), reflected by a percentage of edema inhibition of 62.52% and 62.01%, respectively. 

#### 2.4.3. In Silico Results

As is shown in Figure 5, the superimposition of the cocrystalized ligand revealed a root mean square deviation of 0.959 Å. The binding affinities of the hit compounds and the MM/GBSA calculation is shown in Figure 6. Eight (8) hit compounds identified from the GC/MS analysis of *M. communis* showed high binding scores comparable to the reference drug (diclofenac). Two reference drugs were used for the comparison: diclofenac was used as the standard in the in vitro and in vivo studies, and celecoxib was also used, having a higher sensitivity ratio towards COX-2. The chemical structure of selected compounds and drugs is shown in Table 3.

Among the hit compounds, cohumulinic acid, with a docking score of −7.789 kcal/mol, had a higher binding affinity than diclofenac (−7.768 kcal/mol), and this was less than the cocrystalized ligand 2,5-Cyclohexadiene-1,4-dione,2,5-bis(1,1-dimethylpropyl), which had a docking score of −8.106 kcal/mol. Celecoxib (−11.474 kcal/mol) showed a higher binding affinity than all of the hit compounds and diclofenac, as shown in Figure 6. The MM/GBSA result revealed the same trends as those observed in the docking score, with celecoxib having a higher binding free energy (−60.437 kcal/mol) than all of the hit compounds. Among the top-scoring compounds, 5-Isopropyl-2,2,7a-trimethyl hexa hydro benzo[1,3]dioxol-4-ol; 28237-β-Selinene showed the highest binding free energy (−38.1 kcal/mol), which was even more than for diclofenac (−13.7 kcal/mol) but less than for the cocrystalized ligands (−44.9 kcla/mol).

Table 3 revealed that, for most of the hit compounds, the reference drug and cocrystalized ligand formed one (1) hydrogen bond interaction with SER 516. This was present at the binding site of COX-2. Moreover, 4-Hexen-1-ol, 6-(2,6,6-trimethyl-1-cyclohexenyl)-4-methyl-, (E) formed one H-bond with TYR 341, while Grandlure II formed an H-bond with TYR 371. Some of the hit compounds did not form H-bonds but formed other forms of interactions, as revealed in Figure 7. The 2D representation of all of the interactions between the functional groups of the top-scoring compounds and the amino acid residues at the binding site of the target, with H-bond and pi–alkyl interactions being the major observed interactions, is presented.

## 3. Discussion

Medicinal plants have long been employed in folk medicine as an alternative source of pharmaceuticals, especially anti-inflammatory medications. One of the most serious problems with employing medicinal plants in medicine is a lack of experimental data on their efficacy. In this study, we present the experimental results regarding the chemical composition and antioxidant and anti-inflammatory activities of essential oils extracted from *Myrtus communis* L. leaves.

### 3.1. Phytochemical Analysis

According to the bibliography, the total essential oil of plants is low and rarely above 1% by mass [1,38]. In this study, the yield of myrtle EO was reasonable compared with those reported in the literature. For example, the yield of the aerial parts harvested in the Gouraya region (the northeast of Algeria) was 0.77% [39]. The yield was less than that of *M. communis* grown in Palestine (1.31 and 1.15% in Jericho and Jenin, respectively) [40]. The average yield of myrtle from the Bainem forest (northwest of Algiers) was 0.33% [27]. Indeed, variability in results was also documented in the EO extracted from myrtle leaves in other studies; the recorded yield varied from 0.33 to 0.74% in Portugal [41], 0.90% in Italy [31], 0.35% in Iran [42], 0.48 to 0.80% in Albania [43], and 0.72 to 0.82 in Montenegro [44].

GC–MS analysis showed a predominance of oxygenated monoterpene compounds in MCEOs, as shown in Table 2. These findings support prior data that indicate the presence of 1.8-cineol as one of the main constituents of myrtle EO, as well as α-pinene [27,45]. Notwithstanding the extraction process, α-pinene and 1,8-cineole are the predominant components in Algerian myrtle leaf essential oils, correlating with French and Tunisian myrtles [46,47]. Another myrtle EO, from Iran, showed that α-pinene, 1,8-cineol, limonene, linalool, α-terpineol, and lynalyl acetate were the predominant components, with a marked absence of myrtenyl acetate [48].

According to the presence/lack of myrtenyl acetate, it was feasible to hypothesize two chemotypes of *M. communis* [49]. Since myrtenyl acetate was absent, our oil sample seemed to belong to the second chemotype, i.e., α-pinene/1,8 cineole, proposed by Bradesi et al. [45]. Our α-pinene/1,8 cineole chemotype was vividly confirmed by a study carried out on myrtle from central northern Algeria [50] and by other studies [51,52].

Thus, the remarkable fluctuations in the EO yield and variability in its chemical composition, even among the same specimens, could be due to the distillation method and analysis conditions [27], or many extrinsic factors like temperature and rainfall [53], the geographic source of the species and the season of harvest [52,54], the stage of ripening, storage conditions of the sample (exposure to light, relative humidity, etc.), and genetic and growth factors [53,55,56].

### 3.2. Antioxidant Activity of MCEOs

Studying the antioxidative capacity of natural products could be considered as the first stage for screening new alternatives for therapeutic drug development and understanding the multifaceted scenario of their biological behavior in different diseases [57]. In terms of providing hydrogen ions or electrons by phenol group (-OH), the antioxidant activity of MCEOs was estimated by DPPH, ABTS, and hydroxyl radical scavenging assays. EOs from myrtle exhibited a high free radical scavenging ability and reducing potential when compared to gallic acid as a proven pure antioxidant compound. In the DPPH free radical scavenging test, myrtle showed an excellent scavenging ratio as high as 75.79 ± 0.56% with an IC50 of 15.317 ± 0.340 µg/mL compared to gallic acid. Consistently, MCEOs exhibited a strong ABTS and hydroxyl radical scavenging ability, reflected by an IC50 of 18.890 ± 2.190 µg/mL and 31.877 ± 0.742 µg/mL, respectively, compared to the standard antioxidant. These results are approximately in accordance with the previously published data in which essential oils from Algerian myrtle showed the highest antioxidant capacity, with an IC50 of 45.76 μg/mL, a stronger β-carotene bleaching inhibition capacity, and a remarkable ability to reduce iron (Fe) at a concentration of (60 μg/mL) [32]. The higher scavenging ability of MCEOs is supported by previous studies [58], and it can be attributed to the values in terms of terpenes with conjugated double bonds and oxygenated monoterpenes dominated by 1,8-cineole in this oil [59].

In a dot blot test, methyl eugenol and 1,8-cineole were chosen as powerful scavengers due to the phenolic or aromatic moieties in their molecular structures [44]. Snoussi et al. [58] reported that eugenol and methyl eugenol are responsible for a 1.1% and 0.6% contribution to the total oil and play an important role in antioxidant capacity. In the study conducted by Lim et al. on lemon myrtle from the Myrtaceae family, a considerable DPPH scavenging activity was found with an IC50 of 42.57 µg/mL, mostly ascribed to oxygenated monoterpenes [59]. In the same line of research, Snoussi et al. [58] declared a remarkable scavenging ability of essential oils from Tunisian myrtle with an IC50 of 240 ± 2.90 μg/mL, probably due to the presence of 1,8-cineole. Hennia et al. [60] stated that MCEOs at different concentrations have poor to moderate antioxidant activity, linked to the high amount of α-pinene and β-pinene. The change in antioxidant activity can be explained by the variation in the chemical composition of the EOs, depending on the tested concentrations, variability of phenolic compounds, and structure–activity relationship [61]. It is not easy to precisely determine how EOs act as antioxidants due to the intricacy of their multiple components and the lack of studies on their molecular mechanism. Sometimes, in addition to the major compounds, minor compounds, even in small quantities, are decisive in inducing a biological antioxidant activity [62]. This indicates a possible synergistic interaction between the constituents [63]. However, the association between major and minor compounds may lead to an antagonistic effect [64,65,66,67].

### 3.3. Anti-Inflammatory Activity

Since oxidative stress and inflammation are closely interrelated [68,69,70], it was therefore necessary to study the anti-inflammatory potential of the oil sample.

For in vitro study, denaturation of protein is the process by which proteins lose their secondary and tertiary structures under external conditions [71]. It is well-documented that denaturation of tissue proteins is an important source of inflammation [13,72]. The antidenaturation activity observed for BSA when interacting with biologically active compounds could be considered to be the first insight in understanding their pleotropic effects and detecting new, effective alternatives to be introduced in novel drug formulations [73]. MCEOs were effectively compared to the standard drug, that is, diclofenac sodium. This was concluded by comparing their IC50 average values. MCEOs possessed an IC50 value of 60.351 ± 5.832 µg/mL, whereas that of diclofenac sodium was found to have a value of 7.110 ± 0.624 µg/mL. Our findings showed a good level of protection against heat denaturation of the protein after a preliminary screening with the BSA assay. Inhibition of BSA denaturation reached over 64%. This result agrees with those of other, previous studies involving myrtle EOs, where the highest percentage inhibition of the BSA denaturation resulted in approximately 64.08% at 200 μg/mL [32]. The authors ascribed this to the presence of oxygenated monoterpenes such as eucalyptol and linalool and their multiple overlapping mechanisms, either singly or in combination. A possible interaction/binding site of molecules with an antidenaturation effect on BSA was predicted based on one- and two-dimensional proton nuclear resonance (1D and 2D 1H NMR) assignments [73]. It was suggested that bioactive molecules may interact with two interesting binding sites localized in the aromatic tyrosine-rich and aliphatic threonine, around the lysine residue on the BSA. This activates the tyrosine-motif-rich receptor responsible for the regulation of the signal transduction biological pathway to ensure their overall biological action [73].

For in vivo study, CAR-induced paw edema in rats is a routine model used in the screening of new anti-inflammatory molecules [74]. Due to the irritant nature of CAR, its intraplantar injection causes a significant increase in paw diameter in all groups. This result is supported by similar previous studies [75]. The mechanisms underlying this model have not yet been fully elucidated [76]. It has been reported that CAR produces an acute inflammation event encompassing a biphasic episode [77]; the early phase is attributed to a release of inflammatory mediators such as histamine, serotonin, and bradykinin [78]; the late phase is due to a release of pro-inflammatory mediators such as PG-like substances [68]. However, during the first hour, the reference drug and the EO at two dose levels did not inhibit paw edema. This finding was in agreement with those of other researchers [79]. In the last two hours (late phase), significant improvements were observed in groups treated with the two tested concentrations of MCEOs and the standard drug. This situation causes us to suggest many hypotheses that need to be confirmed and validated on the molecular level. Like other NSAIDs, diclofenac (DCF) works by inhibiting the biosynthesis of PGs [80]; the aforementioned result can be explained by the fact that PGs were absent in the early phase. The EO, especially at the lowest concentration (25 µg/mL), demonstrated a good anti-inflammatory activity on the CAR-induced paw edema, which proves that our oil sample is more effective at very low doses. These data were consistent with the findings of other researchers [36,52]. The anti-inflammatory activity of DCF is attributed to its capacity in mimicking COX-2 enzyme activity that converts arachidonic acid to PGs. Therefore, NSAIDs inhibit only the late phase where PGs and COX-2 enzymes are detectable. It can be suggested that our MCEOs exhibit anti-inflammatory properties, traceable to the capacity of inhibiting the activity of COX-2 enzymes, just like the reference drug.

The GC–MS analysis revealed the presence of several structurally diverse, bioactive chemical constituents in the EO. The most accredited hypothesis may be that the bioactive constituent might be responsible for the anti-inflammatory efficacy of the oil. Several studies have outlined the biological anti-inflammatory activity and analgesic effect of alpha terpineol tested individually; therefore, the species containing this compound are used as expectorants and diuretics as well as for the management of muscle pain relief [81]. Due to the phytocomplexity of EOs, recognizing their mechanism of action is often tricky [82]. It is difficult to identify just one pathway of molecular action. It seems that each of the constituents of the EO has its own mechanism of action or acts in a synergic or additive way. It has been vividly proven that major compounds of the EO from *Cinnamum osmophloeum* were not responsible for its excellent anti-inflammatory effect [83], which suggests that minor constituents or synergistic effects among the components are responsible for the beneficial effect of the EO tested [65,83]. Our study adds interesting information regarding the anti-inflammatory potential of essential oils from Algerian *Myrtus communis* L., demonstrating their effectiveness in inhibiting the COX-2 enzyme based on a molecular docking analysis; in this analysis, nine compounds presented affinity with the ligand. Identifying, isolating, and purifying the active components responsible for the observed biological activity would be worth exploring because they have the advantage of being natural-based products since they come from natural sources [78].

### 3.4. In Silico Analysis

Virtual screening of bioactive compounds from natural sources enables the prediction of molecular interaction between the binding domain of the target protein and the functional groups of the compounds by molecular docking. The selection of hit compound(s) for validation as a drug candidate is also enhanced by this analysis. The binding energy of the protein–ligand(s) interaction is used to expressed the binding affinity [84].

In this study, the molecular interaction of sixty-three (63) compounds obtained from GC–MS analysis of *Myrtus communis* L. was studied within the binding site of cyclooxygenase-2. This was carried out in order to estimate their anti-inflammatory potential using an in silico approach as a confirmatory model to the in vitro and in vivo experiments in this study. Before carrying out molecular docking on the bioactive compounds, validation of the molecular docking procedure was performed by removing the ligand cocrystalized with the protein and redocking into the same binding pose after preparation. The redocked ligand slightly deviated from the original geometry (RMSD = 0.959 Å), as shown in Figure 5.

After careful analysis of the docking result using the SP and XP filtering precision, nine compounds were observed to have high binding energy as compared with the reference drugs (diclofenac and celecoxib). As is shown in Figure 6, two hit compounds (97456 = −8.106 kcal/mol and 5365821 = −7.789 kcal/mol) were observed to have a higher binding affinity above the reference drug (diclofenac = −7.768 kcal/mol). The more negative the docking score, the stronger the interaction and binding affinity [85]. The result agrees with the findings of Omoboyowa [84], who reported the inhibitory activity of phytochemicals from *Jatropha tanjorensis* leaves using a computational approach. Celecoxib, a more selective drug for COX-2, was observed to have a higher docking score (−11.474 kcal/mol) than diclofenac and all of the hit compounds.

Molecular docking is a widely accepted model for the virtual screening of small molecules against a target protein, although it lacks vital parameters for energy estimation. Hence, the molecular mechanics with generalized Born surface area (MM/GBSA) calculation presents the accurate binding free energy (ΔGbind) of the protein–ligand interaction [86]. Therefore, the MM/GBSA results of the hit compounds–cyclooxygenase-2 complexes are provided in Figure 6. The prime MM/GBSA (ΔGbind) of 97456 (−44.890) was observed to be the highest, followed by 540492 (−38.117). All of the hit compounds except 565273 and 86895 were observed to have a better MM/GBSA compared with the reference drug (diclofenac = −13.727).

The interaction of bioactive compounds with the amino acid residues at the binding site of the target protein is important for their reported antagonistic property [87]. Table 4 and Figure 7 show the various interactions between the hit compounds and the amino acid residues at the binding site of cyclooxygenase-2 (3NTG), which might contribute to the binding affinity of the hits–3NTG interaction observed in this study. From Figure 7, Van der Waals, pi–sigma, conventional hydrogen bond, pi–sulfur, amide–pi stacked, and pi–alkyl interactions were observed between the hit compounds and the amino acid residues of the binding site. All of the hit compounds except 28237, 565273, 5281520, and 86895 showed one conventional hydrogen bond interaction with either serine at position 516 or tyrosine at position 341 or 371 within a distance of 5 Å. This conventional H-bond contributes significantly to the binding affinity of the protein–ligand complex as observed in this study. Celecoxib exhibited the highest H-bond interaction (five H-bonds) compared with the hit compounds, cocrystalized ligand, and diclofenac (Table 3). Olawale et al. [88] reported that the favored binding energy of natural compounds results from the H-bond interaction between the functional group of the ligands and amino acid of the target protein. Hence, the presence of the H-bond interaction, as shown in Figure 7, might contribute to the binding affinity of the hit compounds.

## 4. Materials and Methods

### 4.1. Chemicals and Reagents

The analytical-grade reagents and solvents used in this study were obtained from a registered supplier. Furthermore, 1,1-diphenyl-2-picrylhydrazyl (DPPH) and ABTS (2,2′-azino-bis (3-ethylbenzothiazoline-6-sulfonic acid)) free radicals and carrageenan were purchased from Sigma-Aldrich, St. Louis, MI, USA.

### 4.2. Collection of Plant Material

Freshly harvested plant materials (myrtle leaves) were randomly collected (in the early morning) in April 2022, from Taxana, Jijel, Algeria, and identified as *Myrtus communis* L. The region is located 22 km southeast of the city of Jijel (latitude: 36°39′38″; longitude: 5°47′28″; altitude: 750 m).

### 4.3. Essential Oil Extraction

Fresh leaves were collected, cleaned, and ground to obtain a total weight of 100 g. The powdered leaf was subjected to hydro-distillation for three hours with distilled water (500 mL) using a Clevenger-type apparatus according to the standard procedure described by the European Pharmacopoeia (2016). The EO was then separated and kept in glass vials, sealed, and stored in a refrigerator (+4 °C) until further analysis. The yield of extraction was expressed as a percentage and was calculated as follows:$$Extract\;yield\;(\%)=\frac{Mass\;of\;essential\;oil}{Mass\;of\;plant\;material}\times 100$$

### 4.4. Gas Chromatography–Mass Spectrometry (GC–MS) Analysis of EO

The profiling of chemicals present in the oil was performed by a coupling GC–MS system to identify and quantitatively analyze individual components. A volume of 1.0 µL of the working reagent was aspirated into the spectrophotometer in the split mode with 20 (%) ratios on a Shimadzu QP2010, EI 70ev, silica gel capillary column OV1701 (25 m, 0.25 µm) mass spectrometer. The temperature of the column was initially set at 60 °C for 5 min and gradually taken to 240 °C. The analysis was performed for 60.33 min. Helium (99.9999% purity) was used as the gas carrier with a flow rate of 1.0 mL/min. Retention time (Rt) with respect to their spectra was involved in identifying the compounds in comparison with the NIST05 2010 library. The retention index (RI) of the molecules was obtained from http://www.flavornet.org/f_kovats.html (accessed on 11 February 2022) [37].

### 4.5. Antioxidant Activity

#### 4.5.1. Scavenging Activity of the DPPH• Radical Assay

The antioxidant capacity of the oil was estimated by the scavenging ability of free radical DPPH• according to the method of Hatano et al. [89]. The EO was diluted with ethanol to prepare sample concentrations (0–40) µg/mL. Briefly, 0.5 mL of EO was added to an aliquot of 1.5 mL (0.2 mM) of DPPH• dissolved in ethanol. This was mixed thoroughly and allowed to stand at 25 °C for 30 min. Absorbance was recorded at 517 nm. Gallic acid acted as a positive control. The assay was performed in triplicate, and the DPPH• scavenging capacity was evaluated as follows:$$\%\;DPPH\;scavenging\;activity=\frac{Absorbance\;of\;control-Absorbance\;of\;sample\;}{Absorbance\;of\;control}\times 100$$

#### 4.5.2. Hydroxyl Radical (OH•) Scavenging Assay

The OH• scavenging potential of the oil was implemented according to the approach described by Kutlu et al. [90]. Briefly, 1 mL of EO/gallic acid at different concentrations was added to 1.5 mM FeSO_4_ (1 mL), 6 mM H_2_O_2_ (0.7 mL), and 20 mM sodium salicylate (0.3 mL). After 1 h of incubation at 37 °C, the cocktail’s absorbance was read at 562 nm. The same formula used for the DPPH experiment was selected for assessment of the hydroxyl radical neutralizing capacity.

#### 4.5.3. Scavenging ABTS•+ Radical Test

The 2,2′-azino-bis(3-ethylbenzothiazoline-6-sulfonicacid) (ABTS) scavenging activity of the oil was evaluated spectrophotometrically, as described by Re et al. [91] with slight changes. Briefly, the radical was generated through the reaction between 7 mM of ABTS and 2.45 mM of potassium persulfate in distilled water and stored for 24 h in a dark room. Ethanol was used to adjust the concentration of ABTS•+ to give an absorbance of 0.700 ± 0.02 at 734 nm. Briefly, 1.5 mL of the ABTS•+ solution was blended with 50 µL of EO/gallic acid at different concentrations (0 to 40 μg/mL). After incubation in the dark for 10 min, the absorbance of each sample was measured at 734 nm. The results were expressed as percentage inhibition (PI), calculated according to abovementioned formula for the DPPH assay.

### 4.6. Anti-Inflammatory Activity

#### 4.6.1. In Vitro: Inhibition of BSA Denaturation Test

The in vitro anti-inflammatory potential of MCEOs was performed by the protein denaturation inhibition method as described in [92] with a few modifications. The test consists of preparing the reaction mixture containing 2.5 mL of PBS (pH = 6.4), 0.5 mL of BSA (5%) solution, and 50 μL of the EO at different concentrations (12.5–100μg/mL). A control was prepared under the same conditions with sodium diclofenac (reference standard). The mixture was allowed to stand at 37 °C for 15 min and was then reincubated at 70 °C for 5 min. After cooling, the absorbance was read at 600 nm. The percentage protein denaturation inhibition was estimated as follows:$$\%\;inhibition=\frac{Absorbance\;of\;control-Absorbance\;of\;tested\;sample}{Absorbance\;of\;control}\times 100$$

#### 4.6.2. In Vivo: Carrageenan-Induced Paw Edema in Rats

A total of twenty-four white female albino Wistar rats, weighing between 100 and 200 g, were purchased from the Pasteur Institute (Algeria). The animals were kept in polypropylene cages and adapted under standard environmental conditions of 22–24 °C with a relative humidity of 55 ± 10% and a 12 h:12 h light/dark cycle daily. All animal groups had free access to standard pellets and water ad libitum during the acclimatization period. All experimental assays in this study were performed based on the internationally accepted protocols and ethics for laboratory animal care and use, as approved by the committee of the Algerian Association of Sciences in Animal Experimentation (N°. 8808/1988), associated with veterinary medical activities and animal health protection (N° JORA:004/1988).

The in vivo anti-inflammatory activity was assessed according to the method of Winter et al. [93] with slight modifications. Tested animals were randomized into four groups of six animals each (n = 6). The groups and associated interventions are presented in Table 4. Before testing, rats were fasted for 18 h with free access to water. A single dose of freshly prepared diclofenac (50 mg/kg) was used as the reference drug due to its well-known anti-inflammatory features, and graded low doses of MCEOs (25–50 µL/kg) were given to rats orogastrically (per oral). Then, 30 min later, the rats received an intraplantar injection (*ipl*) of 100 µL of freshly prepared carrageenan (1% in NaCl 0.9%) suspension into the right hind paw. Paw thickness was measured using a vernier caliper just before starting the induction of inflammation (time 0) and after the injection of carrageenan at hourly intervals for 4 h.

**Table 4 pharmaceuticals-16-01343-t004:** Treatment protocol used to evaluate the ability of MCEOs to reduce carrageenan-induced paw edema in rats.

Groups	Status	Optimum Doses and Route of Administration	Justification of Optimal Doses Choice
Group I	Control group	Vehicle (distilled water) (per os) + 100 µL of carrageenan (1%) (*ipl*)	[93]
Group II	Experimental group A	Diclofenac (50 mg/kg) (per os) + 100 µL carrageenan (1%) (*ipl*)	[94,95]
Group III	Experimental group B	MCEOs (25 mg/kg) (per os) + 100 µL carrageenan (1%) (*ipl*)	[35]
Group IV	Experimental group C	MCEOs (50 mg/kg) (per os) + 100 µL carrageenan (1%) (*ipl*)	[35]

(ipl): intraplantar injection.

The evaluation of edema was studied by estimating the average percentage increase (% AUG) in the paw volume of rats according to the formula:$$\%AUG=\frac{Paw\;volume\;at\;time\;t-initial\;paw\;volume}{initial\;paw\;volume}\times 100$$

Anti-inflammatory activity was estimated by calculating the percentage inhibition (% INH) of edema as follows:$$\%\;INH=\frac{\%\;AUG\;control-\%\;AUG\;treated}{\%\;AUG\;control}\times 100$$

The % AUG control: % AUG in paw volume of control rats given carrageenan alone; % AUG treated: % AUG in paw volume of pretreated rats with MCEOs, tested at different doses.

### 4.7. In Silico Study

#### 4.7.1. Preparation of Cyclooxygenase-2 Crystallographic Structure and Generation of Receptor Grid

The cyclooxygenase-2 structure, PDB ID of 3NTG with a resolution of 2.19 Å, and cocrystalized ligand were downloaded from the research collaborator for the structural bioinformatics protein databank. The target was prepared using the Schrodinger suite (2017, v1), using the protein preparation wizard to ensure all missing hydrogen atoms were replaced, bond orders were assigned, energetic optimization during refinement with an OPLS3 force field was performed, and the RMSD of heavy metals was set at 0.3 Å [86]. The gride glide file was generated using the receptor grid generation tool at the binding site of the cocrystalized ligand with coordinates x = 26.73, y = 21.49, and z = 17.16.

#### 4.7.2. Preparation of Compounds

Fifty-eight compounds from the GC–MS result of oil extracted from *Myrtus communis* L. were downloaded as an SDF from the following website: https://pubchem.ncbi.nlm.nih.gov accessed on 11 February 2022. The compounds were imported using Schrodinger work space and prepared using Ligprep for possible ionization states assigned at a physiological pH of 7.2 ± 0.2 [96].

#### 4.7.3. Virtual Screening Procedure

The virtual screening of the 58 compounds and reference ligands was performed using two out of the three hierarchical GLIDE docking filters, namely, standard precision (SP) and extra precision (XP). XP filtering precision is a more robust procedure that runs for longer than SP [86]. XP is use for screening ligands determined to have high-scoring poses. All of the compounds were subjected to SP docking, and the ten (17.2%) best-scoring compounds by binding affinities were screened using XP.

#### 4.7.4. MM–GBSA Analysis

MM/GBSA is an advanced quantum mechanics estimation that eliminates false-positive results from docking procedures [97]. In this study, XP filtering complexes were minimized by Prime using the OPLS3 force field, and the binding energy (∆^bind^) was determined as follows:$${\Delta G}^{bind}={\Delta E}^{MM}+{\Delta G}^{Solv}+{\Delta G}^{SA}$$

### 4.8. Data Analysis

Data obtained from the laboratory experiment were analysis using one-way analysis of variance (ANOVA) followed by a post hoc LSD test; the level of significance at *p* < 0.05 was accepted using GraphPad Prism 8. All of the results were presented as the mean ± standard deviation.

## 5. Conclusions

In conclusion, the gathered outcomes herein are intriguing and show that EOs isolated from leaves of Algerian *Myrtus communis* L. are rich in a plethora of key phytochemicals. Biological evaluation revealed that a broad spectrum of antioxidant and anti-inflammatory features is encrypted in MCEOs. Similar antioxidant activity to gallic acid, in a dose-dependent way, was observed. The binding affinity and molecular interaction of the hit compounds from *Myrtus communis* L. against cyclooxygenase-2 suggest its anti-inflammatory potential, corroborating the observed in vitro and in vivo anti-inflammatory activities. Out of 60 identified molecules, 9 minor compounds demonstrated the best stability and interactions as compared to diclofenac. In light of these findings, our EO sample could be a promising candidate and a proposed tool that may provide new insights into the field of drug design and the cosmetic, agricultural, and foodstuff industries. In terms of future perspectives, the transition of in vitro experiments to long-term in vivo trials to validate the efficacy of MCEOs in the treatment of chronic diseases still poses a challenge. Therefore, further research in nonhuman primates would be worth exploring.

## Figures and Tables

**Figure 1 pharmaceuticals-16-01343-f001:**
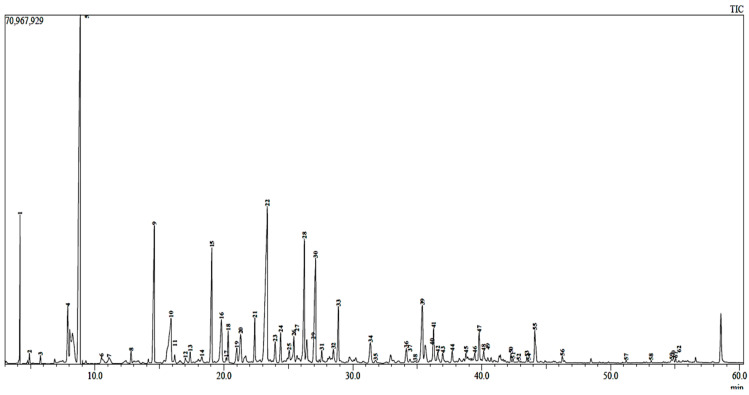
Representative chromatogram of *Myrtus communis* L. essential oil extracted by steam distillation method.

**Figure 2 pharmaceuticals-16-01343-f002:**
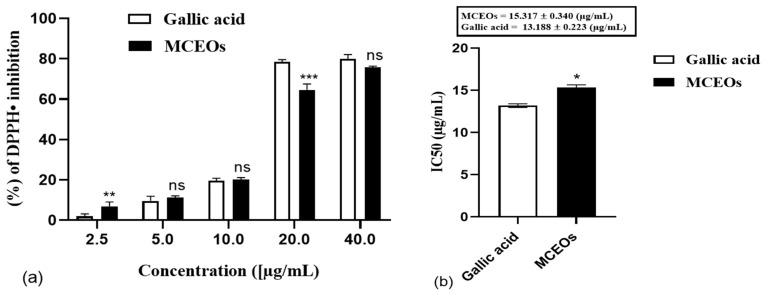
Antioxidant properties of MCEOs and gallic acid (as positive control) through DPPH (**a**), ABTS (**c**), and hydroxyl radical scavenging assays (**e**), and their IC50 (**b**,**d**,**f**), respectively. Data are expressed as mean ± S.E.M (n = 3). ns: no significant difference, *: *p* < 0.05, **: *p* < 0.01, ***: *p* < 0.001 (*t* test) using GraphPad Prism 8 for Microsoft Office.

**Figure 3 pharmaceuticals-16-01343-f003:**
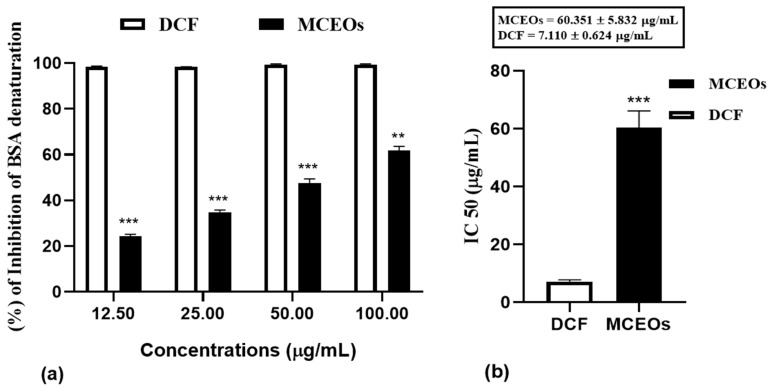
(**a**) inhibition of heat-induced BSA denaturation by *MCEOs*. Data are expressed as mean ± S.E.M (n = 3). (*t* test: (*p* < 0.05) Mean values of samples showing significant difference compared to the control (untreated 5% BSA water solution), **: *p* < 0.01, ***: *p* < 0.001). (**b**) IC50 analyses performed using GraphPad Prism 8 for Microsoft Office.

**Figure 4 pharmaceuticals-16-01343-f004:**
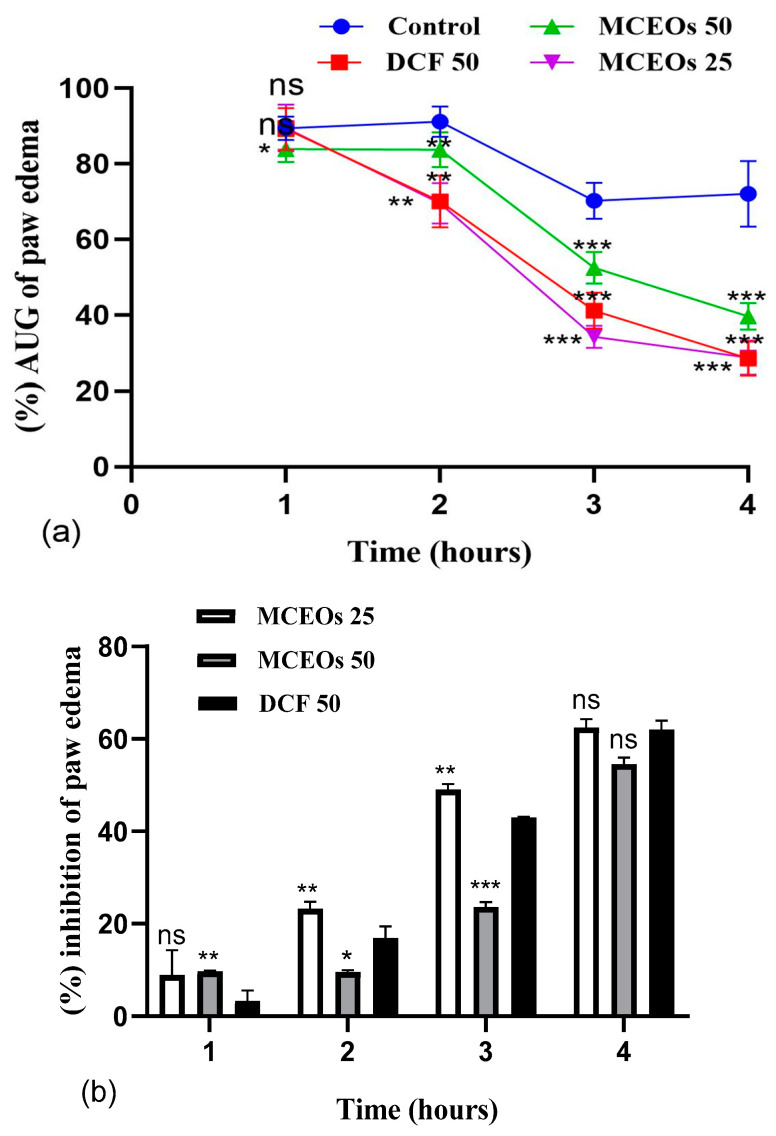
(**a**) Influence of MCEOs (25–50 mg/kg) on CAR-induced paw edema. Data represent the percentage augmentation of paw edema (mean ± SEM) in different groups. (**b**) Percentage inhibition of paw edema in rats treated with the essential oil of *Myrtus communis* L. (MCEOs) or diclofenac (DCF). Data are expressed as mean ± S.E.M (n = 6). ns: no significant difference, *: *p* < 0.05, **: *p* < 0.01, ***: *p* < 0.001.

**Figure 5 pharmaceuticals-16-01343-f005:**
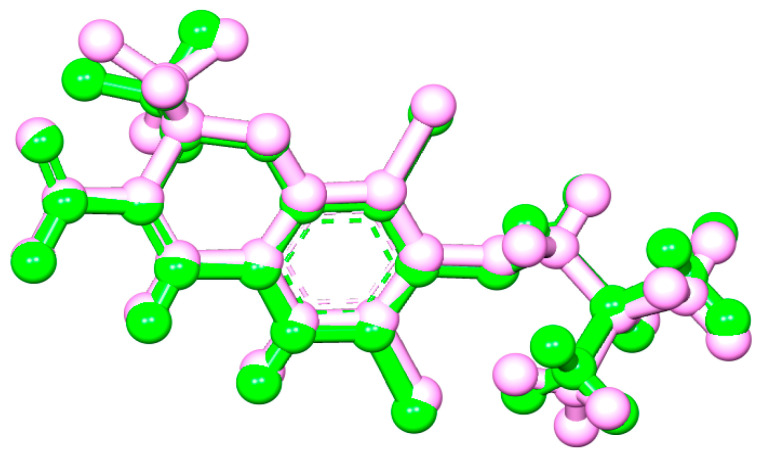
Cyclooxygenase-2 (3NTG) cocrystalized ligand superimposed with its docked pose (RMSD = 0.959 Å).

**Figure 6 pharmaceuticals-16-01343-f006:**
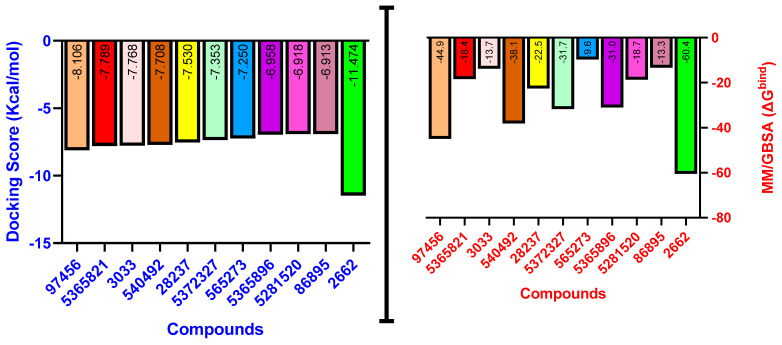
Graphical representation of the molecular docking score and Prime/MM–GBSA binding energy (Δgbind) of hit compounds and reference ligand: 97456—2,5−Cyclohexadiene−1,4−dione, 2,5−bis(1,1−dimethylpropyl); 5365821—cohumulinic acid; 540492—5-Isopropyl−2,2,7a−trimethyl hexa hydro benzo [1,3] dioxol-4-ol; 28237—β-Selinene; 5372327—4-Hexen-1-ol, 6− (2,6,6−trimethyl−1−cyclohexenyl)−4−methyl−(E)−565273—3−Isopropyl−6,7−dimethyltricyclo [4.4.0.0(2,8)]decane−9,10−diol; 5365896—Grandlure II; 5281520—Humulene; 86895—(+)−Cuparene; 3033—diclofenac (reference drug); 2662—celecoxib (reference drug).

**Figure 7 pharmaceuticals-16-01343-f007:**
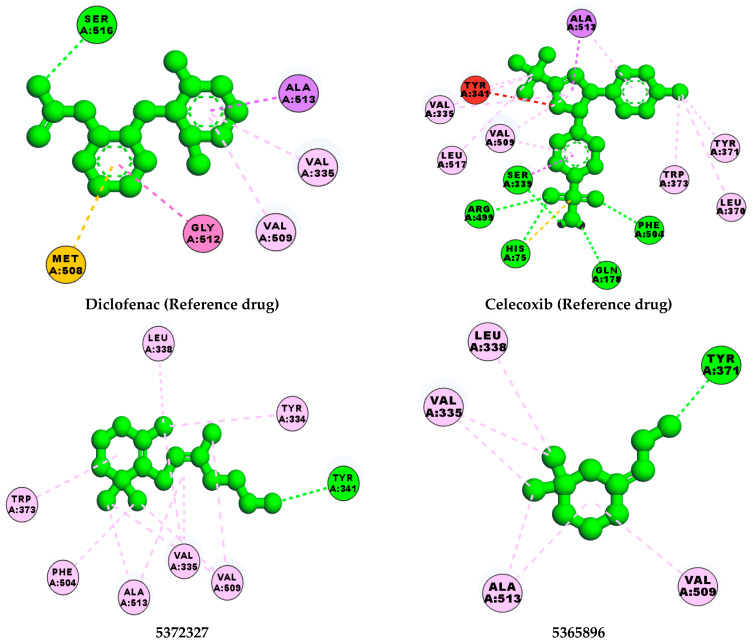
Two-dimensional molecular interaction of hits and reference ligand with amino acids at the binding site of cyclooxygenase-2 viz 5372327—4-Hexen-1-ol, 6-(2,6,6-trimethyl-1-cyclohexenyl)-4-methyl-, (E); 5365896—Grandlure II; 5365821—cohumulinic acid.

**Table 1 pharmaceuticals-16-01343-t001:** Yield and density of essential oils (EOs) obtained by steam distillation of Myrtus communis L. fresh plant material.

Plant Species	Oil Weight (g)	Plant Weight (g)	Yield (%)
*Myrtus communis* L.	0.9	300	0.3

**Table 2 pharmaceuticals-16-01343-t002:** Chemical composition of *M. communis* L. essential oil identified by GC–MS. Rt, retention time; RI, retention index; NF, not found.

Compounds	Percentage (%)	Rt	RI	Base *m*/*z*
1. α-pinene	1.81	4.180	945	93.10
2. β-pinene	0.22	5.764	994	93.10
3. β-myrcene	4.40	7.888	1020	68.05
4. D-Limonene	2.08	8.846	1056	43.00
5. 1,8-Cineol	19.05	10.50	1087	93.10
6. Bicyclo [4.1.0] hept-2-ene, 3,7,7-trimethyl-(2-Carene)	0.24	11.088	1095	93.10
7. cis-p-Mentha-2,8-dien-1-ol	0.40	12.795	1134	70.10
8. Butanoic acid, 2-methyl-, 2-methylbutyl ester	0.40	14.597	1187	93.10
9. β-Linalool	5.70	15.893	1202	69.10
10. (-)-Cis-Sabinol	0.20	16.183	1271	92.05
11. 4-Terpineol (p-Menth-1-en-4-ol)	0.42	17.011	1273	91.05
12. 1,4-Benzodioxan-2-ylmethyl 2-furoate	0.34	17.377	NF	71.05
13. p-Menth-1-en-8-ol (α-Terpineol)	4.62	18.288	1300	95.10
14. p-Menth-1-en-8-ol, acetate (α-Terpinyl-Acetate)	2.50	19.063	1347	59.05
15. Dimethylbenzylcarbinyl Acetate	0.17	19.801	NF	93.05
16. Linalyl acetate	1.04	20.177	1352	132.10
17. (1S-(1Alpha,2alpha,4beta))-1-isopropenyl-4-methyl-1,2-cyclohexanediol	0.77	20.326	NF	93.10
18. Geraniol	1.53	20.994	1367	43
19. *Trans*-Pinocarveol	0.32	21.292	1375	93.10
20. Geranyl Acetate	11.74	22.396	1473	69.10
21. 3-Isopropenyl-5-methyl-1-cyclohexene	1.13	23.363	1489	69.10
22. Eugenol	0.50	23.975	1500	93.10
23. α-Patchoulene	1.22	24.404	NF	93.10
24. γ-Selinene	0.33	25.064	1546	164.10
25. β-Caryophyllene	1.67	25.426	1556	135.10
26. α-Humulene	0.95	25.671	1558	105.10
27. nerol- Acetate	5.07	26.242	1578	69.1
28. Ingol 12-acetate	0.89	26.427	NF	139.10
29. Eugenol Methyl ether	5.58	27.108	NF	178.10
30. 2-Ethyl-5-n-propylphenol	0.48	27.591	NF	135.10
31. 2,4,4-Trimethyl-3-(3-methylbuta-1,3-dienyl)cyclohexanone	0.69	28.492	NF	139.10
32. (E)-Methyl isoeugenol	2.24	28.876	NF	178.10
33. Widdrol hydroxyether	1.35	31.369	NF	139.10
34. Viridiflorol	0.17	31.77	1845	58.00
35. Cohumulinic acid	0.83	34.147	NF	252.10
36. Bicyclo [4.3.0] nonan-2-one, 8-isopropylidene-	0.23	34.44	NF	178.10
37. 2-Hydroxy-3,5,5-trimethyl-2-cyclohexen-1-one	0.22	34.846	NF	173.10
38. Durohydroquinone	3.64	35.398	NF	166.10
39. Caryophyllene oxide 1664	1.24	35.627	1966	41.05
40. Cinerolone	1.45	36.264	NF	166.10
41. Androstan-17-one, 3-ethyl-3-hydroxy-, (5. *alpha*.)	0.73	36.591	NF	41.05
42. 1-Heptatriacotanol	0.60	36.982	NF	41.05
43. 2-Dodecen-1-yl (-) succinic anhydride	0.62	37.718	NF	41.05
44. Furan, 2,3-dihydro-2,2-dimethyl-3-(1-methylethenyl)-5-(1-methylethyl)-	0.40	38.776	NF	43.00
45. Z-5,17-Octadecadien-1-ol acetate	0.58	39.456	NF	43.00
46. 1,2,4-Cyclopentanetrione, 3-(2-pentenyl)–	1.09	39.798	NF	180.10
47. 4′-Ethoxy-2′-hydroxyoctadecanophenone	0.46	40.149	NF	180.10
48. 3-Octen-2-one, 3-butyl-	0.28	42.258	NF	43.00
49. Bicyclo [2.2.1]heptan-2-one, 4-hydroxy-1,7,7-trimethyl-, acetate	0.10	42.427	NF	43.00
50. 3-Isopropyl-6,7-dimethyltricyclo [4.4.0.0(2,8)] decane-9,10-diol	0.07	42.868	NF	159.15
51. 6Z-2,5,5,10-Tetramethyl-undeca-2,6,9-trien-8-one	0.17	43.492	NF	83.05
52. 2-Butenoic acid, 2-methyl-, 2-(acetyloxy)-1,1a,2,3,4,6,7,10,11,11a-decahydro	0.05	43.632	NF	83.05
-7,10-dihydroxy-1,1,3,6,9-pentamethyl-4a,7a-epoxy-				
53. (+/−)-Phytone	1.69	44.115	2314	43.00
54. 5-Isopropyl-2,2,7a-trimethylhexahydrobenzo [1,3] dioxo-4-ol	0.26	46.246	NF	43.00
55. Grandlure II	0.07	51.204	NF	93.05
56. 2,5-Cyclohexadiene-1,4-dione, 2,5-bis(1,1-dimethylpropyl)-	0.06	53.141	NF	318.10
57. Androst-1-en-3-one, 17-(acetyloxy)-4,5-epoxy-, (4. beta.,5. beta.,17. beta.)-	0.13	54.719	NF	329.20
58. 4-Norlanosta-17(20),24-diene-11,16-diol-21-oic acid, 3-oxo-16,21-lactone	0.13	54.873	NF	69.05
59. (+)-Cuparene	0.10	55.041	NF	132.10
60. 4-Hexen-1-ol, 6-(2,6,6-trimethyl-1-cyclohexenyl) -4-methyl-, (E)-	0.06	55.293	NF	329.20

Total identified	98.78%			
Monoterpene hydrocarbons	8.75			
Sesquiterpene hydrocarbons	4.17			
Oxygenated monoterpenes	70.56			
Oxygenated sesquiterpenes	3.10			
Other compounds	12.20			

**Table 3 pharmaceuticals-16-01343-t003:** Hydrogen bond interaction of the docked complex and chemical structure of selected compounds and drugs.

PubChem ID	No H-Bonds	Interacting Residues	Chemical Structure of Compounds
97456	1	SER 516	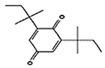
2662	5	PHE 504; GLN 178; HIS 75; ARG 499; SER 339	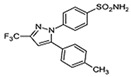
5365821	1	SER 516	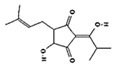
3033	1	SER 516	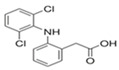
540492	1	SER 516	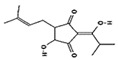
28237	0	Nil	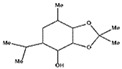
5372327	1	TYR 341	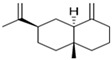
565273	0	Nil	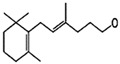
5365896	1	TYR 371	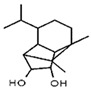
5281520	0	Nil	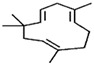
86895	0	Nil	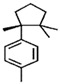

97456—2,5-Cyclohexadiene-1,4-dione, 2,5-bis(1,1-dimethylpropyl); 5365821—cohumulinic acid; 540492—5-Isopropyl-2,2,7a-trimethylhexahydrobenzo [1,3] dioxol-4-ol; 28237—β-Selinene; 5372327—4-Hexen-1-ol, 6-(2,6,6-trimethyl-1-cyclohexenyl)-4-methyl-, (E)-; 565273—3-Isopropyl-6,7-dimethyltricyclo [4.4.0.0 (2,8)] decane-9,10-diol; 5365896—Grandlure II; 5281520—Humulene; 86895—(+)-Cuparene; 3033—diclofenac (reference drug).

## Data Availability

Data is contained within the article.

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
