# Peer review of "Unveiling the Chemical Profiling Antioxidant and Anti-Inflammatory Activities of Algerian Myrtus communis L. Essential Oils, and Exploring Molecular Docking to Predict the Inhibitory Compounds against Cyclooxygenase-2"

_pharmaceuticals, 2023, doi:10.3390/ph16101343_

Round 1

Reviewer 1 Report

The article with the title " Unveiling the Chemical Profiling, Antioxidant and Anti-inflammatory Activities of Algerian’s Myrtus communis L Essential Oils and Exploring Molecular Docking to Predict the Inhibitory  Compounds against Cyclooxygenase-2" aimed to characterize the phytochemical composition M. communis EO, and to evaluate its antioxidant and antiinflammatory activities. The topic of the work could be of interest, but there are several issues and crucial points that need consideration.

-The chemical components have been incorrectly identified. The identification of the chemical composition contains numerous errors. According to the authors, D-limonene has a retention time of 7.888 minutes, while 2-carene has a retention time of 10.52 minutes. This is not possible, because 2-Carene in GC elutes before D-Limonene.

-To determine the chemical composition of a material, use its retention index and fragmentation pattern in the mass spectra. The retention index of a chemical molecule is obtained by interpolating the retention time between neighboring n-alkanes. Retention indices, as opposed to retention periods, allow comparing results determined by different analytical laboratories under varied conditions, whereas retention times vary depending on the individual chromatographic system, such as column length, film thickness, diameter, and inlet pressure.

-The following article should be referenced: M yrtus communis L.: essential oil chemical composition, total phenols and flavonoids contents, antimicrobial, antioxidant, anticancer, and α-amylase inhibitory activity. Al-Maharik, N.Jaradat, N.Al-Hajj, N.Jaber, S. Chemical and Biological Technologies in Agriculture2023, 10(1), 41

-List the identified components in the order of their elution on the SE30 column, along with their retention indices.

-For that reasons, I suggest major revision of the manuscript before to assess it for publication in Pharmaceuticals.

Minor editing 

Author Response

  1. Response to REVIEWER 1

Comment 1: The article with the title " Unveiling the Chemical Profiling, Antioxidant and Anti-inflammatory Activities of Algerian’s Myrtus communis L Essential Oils and Exploring Molecular Docking to Predict the Inhibitory Compounds against Cyclooxygenase-2" aimed to characterize the phytochemical composition M. communis EO, and to evaluate its antioxidant and anti-inflammatory activities. The topic of the work could be of interest, but there are several issues and crucial points that need consideration.

Response 1: The authors appreciate the reviewer’s comment. We would like to extend our sincere appreciation for your time and effort spent on the review of our manuscript.  Your input and expertise are greatly valued, and your suggestions help strengthen the overall quality and robustness of the scientific discourse presented in the article.

Comment 2: The chemical components have been incorrectly identified. The identification of the chemical composition contains numerous errors. According to the authors, D-limonene has a retention time of 7.888 minutes, while 2-carene has a retention time of 10.52 minutes. This is not possible, because 2-Carene in GC elutes before D-Limonene.

Response 2:  After verification of the original analysis report; indeed, it is the cis-carane (Bicyclo [4.1.0] hept-2-ene, 3,7,7-trimethyl-) and not the 2-carene. The name of the molecule has been corrected in Table 2. The authors appreciate the reviewer’s comment.

Comment 3: To determine the chemical composition of a material, use its retention index and fragmentation pattern in the mass spectra. The retention index of a chemical molecule is obtained by interpolating the retention time between neighboring n-alkanes. Retention indices, as opposed to retention periods, allow comparing results determined by different analytical laboratories under varied conditions, whereas retention times vary depending on the individual chromatographic system, such as column length, film thickness, diameter, and inlet pressure.

Response 3:

With respect to your comments about using the retention index to determine the chemical composition of our sample, we completely agree with your notion that using retention indices would increase the reliability and comparability of our results across different laboratories and conditions. The importance of retention indices as an integral part of gas chromatography-mass spectrometry (GC-MS) analysis is well understood.

However, due to the limitations of our laboratory conditions, it was not feasible to determine the retention index. we want to clarify that despite this limitation, we have endeavored to maximize the reliability and validity of our results within the bounds of available resources. Our experiments have been conducted under carefully controlled conditions to ensure consistency, and every attempt has been made to align our procedures and parameters with standard scientific practices.

In order to enhance the comparability of our data, we have used the fragmentation pattern of the mass spectra along with the retention times. Essential oil constituents were identified by comparison of their mass spectra with those stored in the NIST 05 (2010 version, National Institute of Standards and Technology, Gaithersburg, MD, USA) mass spectral library.

Although not as comprehensive as the inclusion of retention indices, this approach still provides valuable insights into the chemical composition of the essential oils under examination.

While we understand and respect the high standards of your esteemed journal, we believe that the limitations we encountered do not undermine the core findings of our study. We hope that you will consider the constraints faced in an academic environment and our earnest efforts to adhere to good scientific practices despite these challenges.

Should you require further clarifications or additional experimental data, we would gladly provide them. Once again, I appreciate your understanding and the constructive feedback provided.   Sincerely

Comment 4: The following article should be referenced: Myrtus communis L.: essential oil chemical composition, total phenols and flavonoids contents, antimicrobial, antioxidant, anticancer, and α-amylase inhibitory activity. Al-Maharik, N., Jaradat, N., Al-Hajj, N., Jaber, S. Chemical and Biological Technologies in Agriculture, 2023, 10 (1), 41.

Response 4: The reference was added.

Comment 5: List the identified components in the order of their elution on the SE30 column, along with their retention indices.

Response 5: identified components were cited in the order of their elution (retention time). m/z values was also added.

Comment 6: For that reasons, I suggest major revision of the manuscript before to assess it for publication in Pharmaceuticals.

Response 6: a major revision of the manuscript was done.

Reviewer 2 Report

The contribution presented by Belahcene et al., is a comprehensive exploration of the essential oil of Myrtus communis and its putative anti-inflammatory effect. While I think the manuscript and its presentation are very good, here are some comments/suggestions that could improve it:

Minor

The resolution of several figures is grainy, I'm not sure if this is due to format or image expansion.

Please consider the addition of a scheme showing the chemical structures of selected compounds.

Diagrams in Figure 7 are not very descriptive, please consider an alternative 2D representation. Similarly, Table 4 does not convey any significant information, description in main text shall be sufficient.

I believe that references are not in the proper format.

Major

There is no comment on the basis for PDBID selection.

The discussion of docking results is poor, the main claim is that proposed ligands showed hydrogen bonding and thus good binders. While hydrogen bonding is indeed very significant not all hydrogen bonds can be weighted as "good". A comparison and overall description of binding site features and significant residues is needed.

The comparison with diclofenac is insufficient. Please consider the addition of another reference compound, with higher selectivity ratio towards COX-2, such as celecoxib.

Finally, while the results do suggest that essential oil components can be good candidates for further evaluation, it is quite the stretch to propose trials in non-human primates. Critical assessment of binding is due prior to further study, in such instances where natural products from essential oils are evaluated it is difficult to limit one compound as "responsible" from the effect. Thus, further testing is needed prior to such scenarios, the conclusion should reflect that.

There are several style errors, the manuscripts needs a through revision and correction.

Author Response

  1. Response to REVIEWER 2

Comment 1: The contribution presented by Belahcene et al., is a comprehensive exploration of the essential oil of Myrtus communis and its putative anti-inflammatory effect. While I think the manuscript and its presentation are very good, here are some comments/suggestions that could improve it:

Response: The authors appreciate the reviewer’s comment. We would like to extend our sincere appreciation for your time and effort spent on the review of our manuscript.  Your input and expertise are greatly valued, and your suggestions help strengthen the overall quality and robustness of the scientific discourse presented in the article.

Minor

Comment 2: The resolution of several figures is grainy, I'm not sure if this is due to format or image expansion.

Response: The resolution of the figures has been improved.

Comment 3: Please consider the addition of a scheme showing the chemical structures of selected compounds.

Response: The chemical structures of selected compounds and drugs has been added in table 3.

Comment 4: Diagrams in Figure 7 are not very descriptive, please consider an alternative 2D representation. Similarly, Table 4 does not convey any significant information, description in main text shall be sufficient.

Response: Figure 7 has been cited in the result interpretation and more descriptive with the relevance of table 4 explained.

Comment 5: I believe that references are not in the proper format.

Response: The references have been corrected according to the journal format.

Major

Comment 6: There is no comment on the basis for PDBID selection.

Response: Information about the choice of the PDB ID has been provided.

Comment 7: The discussion of docking results is poor, the main claim is that proposed ligands showed hydrogen bonding and thus good binders. While hydrogen bonding is indeed very significant not all hydrogen bonds can be weighted as "good". A comparison and overall description of binding site features and significant residues is needed.

Response : The discussion has been improve in accordance to the reviewer’s suggestion.

Comment 8: The comparison with diclofenac is insufficient. Please consider the addition of another reference compound, with higher selectivity ratio towards COX-2, such as celecoxib.

Response: Diclofenac has been included as reference drug in vivo and in vitro and discussed. Celecoxib has been  provided.

Comment 9: Finally, while the results do suggest that essential oil components can be good candidates for further evaluation, it is quite the stretch to propose trials in non-human primates. Critical assessment of binding is due prior to further study, in such instances where natural products from essential oils are evaluated it is difficult to limit one compound as "responsible" from the effect. Thus, further testing is needed prior to such scenarios, the conclusion should reflect that.

Response: Conclusion has been improved in accordance to the reviewer’s suggestion.

Comment 10: Comments on the Quality of English Language. There are several style errors, the manuscripts needs a through revision and correction.

Response: The manuscript has been reviewed by native English speaker.

Round 2

Reviewer 1 Report

Unfortunately, the authors didn’t address my personal concerns. There are many holes in the identification of the EO. The author should seek help in identifying the components of essential oils. I will show a few mistakes that are very clear. The authors claim that myrcene elutes after limonene and linalool. That is not true; the retention index for Myrcene is 985 using the SE30 column, while the RI for limonene is 1027 and for Linalool is 1090. Additionally, Butanoic acid, 2-methyl-, 2-methylbutyl ester (RI = 1096) elutes after linalool, Eugenol (RI = 1360), AND methyl eugenol (RI = 1370) elutes before Beta-caryophyllene Compounds 31 and 34 are methyl eugenol. ?????? The phytochemical analysis is full of mistakes.

Fine

Author Response

Responses to Reviewer 1:

Comments and Suggestions for Authors

The manuscript has some improvements, yet my main concern has not been addressed.

The discussion and treatment of molecular docking results is poor. I understand the main rationale in the discussion. Yet, my initial comment remains valid. Not all hydrogen bonds are significant to binding events. The results from the work clearly show this. Most of the evaluated compounds show interaction with SER516. While celecoxib is the only compound with 5 bonds; yes, but not just that, is the only compound with interactions not shown by others. This is what needs to be adressed and discussed, which of these may or may not be relevant to binding in the context of COX-2 inhibition.

Response : The other interactions observed between the hit compounds and amino acid residues of the active site of COX 2 have been discussed with the conventional hydrogen bond discussed to be the most significant as highlighted in discussion section of the revised Ms.

Similarly, the authors state: "Celecoxib, a more selective drug for COX-2 was observed to have high docking score (-11.474 kcal/mol) than diclofenac and all the hit compounds, this reveals that COX-2 inhibition is the mechanism of action of this drug". The last part is quite troublesome, as it implies severe misconception and faulty logic. Docking scoring is just a number, its validity is at most qualitative in nature. Stating that based on such value X compound has Y mode of action supporting on it is, plain and simple, wrong. Again, the discussion in this matter is the nature of the interaction, yes all of them are hydrogen bonds but which are pharmacophoric and which haptophoric? Are any of these relevant for selective inhibition?

Response : The part ‘this reveals that COX-2 inhibition is the mechanism of action of this drug’ has been removed from the discussion section as highlighted in the revised Ms and convetional H-bond has been discussed has the significant interaction

These are the ideas and answers that the authors must bring to the table. Also, based on the results some of the compounds may exhert some COX affinity indeed, but perhaps it would be biased for COX-1 instead of COX-2 based on the similarity towards the profile shown by diclofenac. This is yet another regard where no comment or mention exists.

Response : COX 1 was not involved in the insilico and in vitro assay, therefore there is no reason to discuss the target that was not involve in the computational study.

Reviewer 2 Report

The manuscript has some improvements, yet my main concern has not been addressed.

The discussion and treatment of molecular docking results is poor. I understand the main rationale in the discussion. Yet, my initial comment remains valid. Not all hydrogen bonds are significant to binding events. The results from the work clearly show this. Most of the evaluated compounds show interaction with SER516. While celecoxib is the only compound with 5 bonds; yes, but not just that, is the only compound with interactions not shown by others. This is what needs to be adressed and discussed, which of these may or may not be relevant to binding in the context of COX-2 inhibition.

Similarly, the authors state: "Celecoxib, a more selective drug for COX-2 was observed to have high docking score (-11.474 kcal/mol) than diclofenac and all the hit compounds, this reveals that COX-2 inhibition is the mechanism of action of this drug". The last part is quite troublesome, as it implies severe misconception and faulty logic. Docking scoring is just a number, its validity is at most qualitative in nature. Stating that based on such value X compound has Y mode of action supporting on it is, plain and simple, wrong. Again, the discussion in this matter is the nature of the interaction, yes all of them are hydrogen bonds but which are pharmacophoric and which haptophoric? Are any of these relevant for selective inhibition?

These are the ideas and answers that the authors must bring to the table. Also, based on the results some of the compounds may exhert some COX affinity indeed, but perhaps it would be biased for COX-1 instead of COX-2 based on the similarity towards the profile shown by diclofenac. This is yet another regard where no comment or mention exists.

Revisions are still needed as style and grammatical errors persist; e.g.

[...] oils from Algerian myrtle showed a highest antioxidant capacity [...]

Author Response

Comments and Suggestions for Authors

Unfortunately, the authors didn’t address my personal concerns. There are many holes in the identification of the EO. The author should seek help in identifying the components of essential oils. I will show a few mistakes that are very clear. The authors claim that myrcene elutes after limonene and linalool. That is not true; the retention index for Myrcene is 985 using the SE30 column, while the RI for limonene is 1027 and for Linalool is 1090. Additionally, Butanoic acid, 2-methyl-, 2-methylbutyl ester (RI = 1096) elutes after linalool, Eugenol (RI = 1360), AND methyl eugenol (RI = 1370) elutes before Beta-caryophyllene Compounds 31 and 34 are methyl eugenol. ??????. The phytochemical analysis is full of mistakes.

Responses :

  1. The authors appreciate the reviewer’s comment. We would like to extend our sincere appreciation for your time and effort spent on the review of our manuscript. Your input and expertise are greatly valued, and your suggestions help strengthen the overall quality and robustness of the scientific discourse presented in the article.
  2. The conclusion has been changed. It is supported by the obtained results.
  3. GCMS results have been verified. All the remarks raised with regard to the molecules elution order are due to a mistake in the name of the capillary column used in our work. We mentioned a false column in the material and methods section. In fact, we used the OV1701 column and not the SE30 (the column name is mentioned in the original report of the analysis).
  4. Regarding the retention indices: due to the limitations of our laboratory conditions, it was not possible to determine the retention index. It was necessary to inject the                    n-alkanes of analytical quality (which are not available in our laboratory) under the same chromatographic conditions to be able to calculate these indices.
  5. Due to this limitation, and to have the kovats indices, we have tried to use libraries like babuschok, adams, ... as well as published papers, however our column no longer appears in these libraries. We found only one library citing our column OV1701 that is http://www.flavornet.org/f_kovats.html. However, it does not contain all the molecules contained in our essential oils.

we remain at your disposal for further details.

Regards

Round 3

Reviewer 1 Report

My issues from the second round of revision were not taken into consideration. Identification of the phytochemical makeup is inaccurate.

I'll only list a few errors.

Before 1, beta myrcene elutes.which elutes before limonene is 8-cineole. 

Before trans-pinocarveol eugenol and b-caryophellene, 4-terpineol elutes.

These are but a few of the errors. Chemists ought to research the phytochemicals.

My issues from the second round of revision were not taken into consideration. Identification of the phytochemical makeup is inaccurate.

I'll only list a few errors.

Before 1, beta myrcene elutes.which elutes before limonene is 8-cineole. 

Before trans-pinocarveol eugenol and b-caryophellene, 4-terpineol elutes.

These are but a few of the errors. Chemists ought to research the phytochemicals.

Author Response

Dear professor,

I would like to thank you for your comments. You find in the following txt the answer of your questions and comments:

  1. The conclusion has been changed. It is supported by the obtained results.
  2. GCMS results have been verified. All the remarks raised with regard to the molecules elution order are due to a mistake in the name of the capillary column used in our work. We mentioned a false column in the material and methods section. In fact, we used the OV1701 column and not the SE30 (The column name is mentioned in the original report of the analysis).
  3. Regarding the retention indices: due to the limitations of our laboratory conditions, it was not possible to determine the retention index. Due to this limitation, and to have the kovats indices, we have tried to use libraries like babuschok, adams, ... as well as published papers, however our column no longer appears in these libraries. We found only one library citing our column OV1701 that is http://www.flavornet.org/f_kovats.html. However, it does not contain all the molecules contained in our essential oils.

I'll only list a few errors.

Before 1, beta myrcene elutes. Corrected

Using OV1701 column: B-myrcene (RI=1020) was eluted after B pinene (RI=994).

which elutes before limonene is 1,8-cineole. 

Using OV1701 column: 1,8 cineole (with RI=1087) was eluted after d-limonene (RI=1056)

Before trans-pinocarveol eugenol and b-caryophellene, 4-terpineol elutes. Corrected.

Using OV1701 column: 4-terpineol (RI=1273) was elutes before b caryophyllene (RI= 1556), eugenol ( RI=1500) and  trans pinocarveol (RI=1375).

These are but a few of the errors. Chemists ought to research the phytochemicals.

GC MS analysis was reviewed.

Comments on the Quality of English Language

We sent it to your English editing MDPI to improve the English.

Best regards

Round 4

Reviewer 1 Report

To the esteemed authors,

As the retention indices for the compounds were not calculated, it is necessary to indicate in the table that the retention indices provided are sourced from the literature.

Good

Author Response

As the retention indices for the compounds were not calculated, it is necessary to indicate in the table that the retention indices provided are sourced from the literature.

Response: Dear Professor, I would like to thank you for your effort and comments.  we putted the references under the table for the retention indices and indicate that is from literature.

Best regards